# IoT and Cloud Computing in Health-Care: A New Wearable Device and Cloud-Based Deep Learning Algorithm for Monitoring of Diabetes

**Ahmed R. Nasser** [1], **Ahmed M. Hasan** [1], **Amjad J. Humaidi** [1], **Ahmed Alkhayyat** [2], **Laith Alzubaidi** [3,*], **Mohammed A. Fadhel** [4,*], **José Santamaría** [5] and **Ye Duan** [6]

1 Control and Systems Engineering Department, University of Technology-Iraq, Baghdad 00964, Iraq; Ahmed.R.Nasser@uotechnology.edu.iq (A.R.N.); 60163@uotechnology.edu.iq (A.M.H.); Amjad.J.Humaidi@uotechnology.edu.iq (A.J.H.)
2 College of Technical Engineering, The Islamic University, Najaf 54001, Iraq; Ahmedalkhayyat85@gmail.com
3 School of Computer Science, Queensland University of Technology, Brisbane, QLD 4000, Australia
4 College of Computer Science and Information Technology, University of Sumer, Thi Qar 64005, Iraq
5 Department of Computer Science, University of Jaén, 23071 Jaén, Spain; jslopez@ujaen.es
6 Faculty of Electrical Engineering & Computer Science, University of Missouri, Columbia, MO 65211, USA; duanye@missouri.edu
* Correspondence: laith.alzubaidi@hdr.qut.edu.au (L.A.); Mohammed.a.fadhel@uoitc.edu.iq (M.A.F.)

**Abstract:** Diabetes is a chronic disease that can affect human health negatively when the glucose levels in the blood are elevated over the creatin range called hyperglycemia. The current devices for continuous glucose monitoring (CGM) supervise the glucose level in the blood and alert user to the type-1 Diabetes class once a certain critical level is surpassed. This can lead the body of the patient to work at critical levels until the medicine is taken in order to reduce the glucose level, consequently increasing the risk of causing considerable health damages in case of the intake is delayed. To overcome the latter, a new approach based on cutting-edge software and hardware technologies is proposed in this paper. Specifically, an artificial intelligence deep learning (DL) model is proposed to predict glucose levels in 30 min horizons. Moreover, Cloud computing and IoT technologies are considered to implement the prediction model and combine it with the existing wearable CGM model to provide the patients with the prediction of future glucose levels. Among the many DL methods in the state-of-the-art (SoTA) have been considered a cascaded RNN-RBM DL model based on both recurrent neural networks (RNNs) and restricted Boltzmann machines (RBM) due to their superior properties regarding improved prediction accuracy. From the conducted experimental results, it has been shown that the proposed Cloud&DL-based wearable approach achieves an average accuracy value of 15.589 in terms of RMSE, then outperforms similar existing blood glucose prediction methods in the SoTA.

**Keywords:** artificial intelligence; deep learning; blood glucose level prediction; type-1 diabetes; cloud computing; IoT

## 1. Introduction

Nowadays, diabetes is considered one of the common diseases that is rapidly increasing in the world. It naturally represents an important health problem in the world and the World Health Organization (WHO) is also encouraging science to work along with this objective. Since 1965, many contributions have been published about promoting specific guidelines for the diagnosis, monitoring, and treatment of diabetes [1]. Regarding diabetes conditions, the human body is unable to create enough insulin to manage blood sugar levels, or the insulin produced is insufficiently utilized. It can also cause various diseases such as diabetes, kidney disease, heart disease, nerve damage, blindness, and damage of blood vessels [2].

In the last few years, one of the most relevant systems used to help in type-1 diabetes (T1D) is continuous glucose monitoring (CGM) systems which allow patients to continuously monitor the blood glucose levels and react based on the current blood glucose levels [3]. Predicting the future blood glucose level can be an early-prediction tool in order to help patients to manage their insulin distribution and protect them from potentially severe damage. However, the complex behavior of blood glucose levels which depends on many factors such as sleeping patterns, recent insulin injections, and carbohydrate intake makes the prediction phase for short terms a complex and challenging task [4]. Thus, it is necessary to adopt those technologies that provide improved alternatives when dealing with these health-care problems.

Recently, artificial intelligence and new learning techniques in the field have become promising approaches for improving the health care of patients. The prediction of glucose levels in the blood can be observed as a time series problem. Therefore, a sequence of present and past glucose levels is used to predict the values of the next levels in the near future. Usually, machine learning (ML) methods are preferred due to they provide more accurate and fast results requiring less computational cost. Deep learning (DL) methods belong to ML, and they allow to increase the predictive feature due to their inherent ability to both combine data from various sources and manage large amounts of data [5].

In the last few years, IoT and Cloud computing paradigms have contributed to the field of health care with new wearable technologies. These cutting-edge developments are crucial for the direct increase of the comfort of the life of the patients dealing with the disease. In particular, instant glucose, glucose trend, and direction information are three relevant parameters that can be obtained by using a small sensor. Such a device requires a low power consumption and it is worn on the arm of the patient, which replaces the traditional invasive blood test. More importantly, these types of IoT devices lack processing power, thus the obtained data can be processed and analyzed in a more powerful computing device placed in the Cloud [6].

In this paper, it is proposed a new wearable CGM system to predict the blood glucose level from glucose level history using a DL method run in the Cloud. Specifically, recurrent neural networks (RNNs) are proven to have the ability to capture temporal auto-correlation features in data, while restricted Boltzmann machines (RBM) have the ability to delineate complex distributions in the data. Therefore, our Cloud&DL-based wearable approach makes use of a DL model designed from the previous learning strategies, and a cascaded RNN-RBM method is accordingly considered. Finally, the new health care system is aimed at time series prediction of the glucose level in the blood for 30 min horizon to provide greater precision than that given by other methods from the state-of-the-art (SoTA).

The contributions can be summarized as follows;

1. A DL-based blood glucose level perdition model is used together with a wearable CGM device to provide T1D prediction of near-future blood glucose levels.
2. A cascaded DL hybrid model based on RNN and RBM to improve the accuracy against the current techniques in the SoTA.
3. A high processing power tool based on using a Cloud computing architecture for running the proposed DL models together with using the lower processing power of a wearable CGM device.

The structure of the paper is organized as follows. A brief summary of the SToA is provided in Section 2. Next, Sections 3 and 4 are devoted to, respectively, describing the proposed approach and providing the details of the system. Section 5 is aimed at performing the experimental evaluation of the proposal. Finally, those more relevant conclusions are introduced in Section 6.

## 2. Review of the State-of-the-Art

This section is aimed at briefly introducing those more relevant contributions in the field. Specifically, blood glucose level prediction is considered an important problem and received considerable attention by researchers to improve the prediction accuracy.

Artificial intelligence approaches such as DL and ML methods are highly considered by the researchers for automatic and accurate prediction of glucose levels in the blood for T1D patients.

In [7], to predict levels of blood glucose 60 min into the future a deep recurrent neural network RNN model is presented. A parameterized univariate Gaussian output distribution is used for estimating the uncertainty in the prediction. They acquire an RMSE of 18.867 using blood glucose level measurements of six people with T1D.

A DL model to predict glucose concentration is presented in [8] using dilated recurrent neural network (DRNN) to achieve a 30-min prediction of the future. The data used are obtained from two sources, both simulator (UVA/Padova T1D simulator) and clinical trials (OhioT1DM). Different inputs such as time index, historical blood glucose, meal intake, and insulin bolus are provided to the network to acquire the glucose level prediction. The result reveals an RMSE of 27.4 mg/dL for the clinical dataset.

ARTiDe presented in [9], which is a model for forecasting blood glucose levels based on a jump neural network JNN to handle both the linear and nonlinear components of inputs data. The model includes temporal delays for the input signals with auto-regressive feedbacks. A private, as well as a public dataset, were used for the predictions of glucose levels of 15, 20, and 30 min in the future the result shows an accuracy of 18.4 RMSE.

A hybrid cascaded DL model is presented in [10] to predict glucose levels in the blood for up to 60 min. The model is based on a single layer from long-short-term memory (LSTM) followed by a bidirectional LSTM layer. The model is trained using different datasets obtained from T1D real patients and simulators. The model has achieved a prediction result with an accuracy of 21.747 in terms of RMSE.

A deep neural network DNN approach is presented in [11] for the state prediction of blood glucose for 30 min in the future in form of (hypoglycemic, euglycemic, and hyperglycemic). The prediction error-grid analysis method is considered to improve the model regarding accuracy. the training dataset is obtained from DirecNet Central Laboratory for 25 T1D patient's time series data. the model has achieved an average of 93% in respect of prediction accuracy.

In [12], a framework called GluNet is presented for forecasting blood glucose levels for T1D patients. The framework is used for CGM forecasting for 30 to 60 min in the future. A deep convolution neural network CNN with label transform/recover method is trained using datasets obtained from Silico virtual adult, Virtual adolescent, and Clinical adult. The model achieved glucose forecasting results of 19.2 RMSE in terms of accuracy.

An LSTM DL method is considered in [13] to model the behavior of blood glucose for T1D patients. The model predicts the blood glucose for 30 to 60 min ahead. A dataset of 5 actual patients with 200 data points is used for training and evaluating the model. The model archived an accuracy of 37.8 regarding RMSE.

In [14], LSTM based DL method is used to predict Blood Glucose Dynamics in respect of three classes (High, Normal, and Low). The model trained with 112 patient data and obtained classification results with an average 86.7% of accuracy.

A comparative study is presented in [15] to evaluate different types of ML and DL methods for glucose level prediction in blood for 30 min ahead. The study considered LSTM DL methods against the classical auto-regression with exogenous inputs ARX. OhioT1DM dataset is used to train and test the performance of the models. Based on the evaluation experiments the ARX method achieved the lowest RMSE of 19.53 compared to the VanillaLSTM DL model which obtained an RMSE of 19.58.

For prediction and forecasting time series data the authors in [16] presented and used a hybrid deep learning model based on a combination of recurrent neural network and restricted Boltzmann machine RNN-RBM. This model is used for different types of time series prediction and forecasting applications such as modeling temporal dependencies in high-dimensional sequence [16], forecasting disasters for the mobile network [17], predicting stock market trends [18], transportation network congestion evolution prediction of network congestion evolution [19], and wastewater treatment plants anomaly predic-

tion [20]. The presented method shows its ability to capture both spatial and temporal dependencies in the data which distinguishes it from other models. RBM is a generative stochastic neural network that learns the probabilistic distribution of the data. Therefore, RBMs have the ability for pattern completion by generating samples based on data distribution. This feature of RBM can be utilized for temporal time series prediction in RNN-RBM architecture where RBM biases are permitted to be adjusted by the RNN to transmit temporal information and RBM parameters are kept constant as a prior for the data distribution.

Different models have their own set of benefits and drawbacks and are better suited to different sorts of tasks and issues, as well as different datasets. In this work, the possible benefits of using the RNN-RBM model to achieve more accurate (compared to SoTA) time-series blood glucose levels prediction will be evaluated.

## 3. The Proposed Approach

This section presents the theoretical framework of the proposed system.

### 3.1. Restricted Boltzmann Machines (RBM)

Restricted Boltzmann machines (RBM) have emerged as one of the most popular probabilistic learning methods. Combined with advances in learning theory, RBMs have expanded their applicability to a variety of tasks such as contrast divergence, persistent and parallel coupling. While successful, most of these models have been used not in the context of relational data, but often with a flat feature representation (vectors, matrices, tensors) [21].

RBM structure shown in Figure 1 has two layers, the first layer is the input layer and the second layer is defined as the hidden layer. Circles have a structure similar to neurons and consist of connections called nodes. Nodes provide connections between layers, but there are no connections between nodes in the same layer.

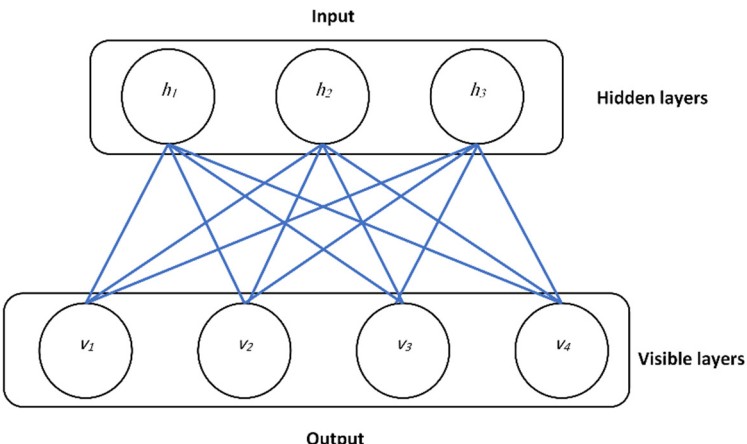

**Figure 1.** RBM Structure.

There is no intra-layer communication. Each node in the layer makes a calculation by making random decisions. All the nodes in the input node layer receive a feature that is lower-level than the item in the dataset that needs to be learned. The x value is multiplied by a weight and added to the bias value at the first node of the hidden layer. The output of the node, or the strength of the signal flowing through it, is produced by feeding the result of these two processes into an activation function that, given input x, creates the output of the node, or the strength of the signal traveling through it. When the combination of the inputs at the hidden node is reviewed, the x values are multiplied by the w weight value, these values are added together and the bias value is added, and the calculated value is passed through an activation function to arrive at the result of the node. The x value for each of the hidden nodes is multiplied by the weight W. Thus, the three weights of each

input x form the sum of the twelve weights. Each hidden node consists of four inputs multiplied by their respective weights. The sum of these values is added to the deviation value and the calculated result value is used with the activation algorithm that produces the result for the hidden node [22].

Equation (1) shows a general form of the energy function for the pair of visible and hidden vectors $\langle v|h \rangle$ the weight matrix $W$ is associated with the connection between $v$ and $h$ of a KBM [23].

$$E(v,h) = -a^T v - b^T - v^T W h \tag{1}$$

here; $a$ and $b$ are the visible and hidden units bias weights. The probability distributions of $v$ and $h$ are expressed with regard to energy function $E(v,h)$ as in Equation (2).

$$P(v,h) = \frac{1}{Z} e^{-E(v,h)} \tag{2}$$

here; Z is the normalizing constant and is given in Equation (3).

$$Z = \sum v', h' e^{-E(v',h')} \tag{3}$$

where $v$ is the probability of vector; It is equal to the sum of Equation (2) over hidden layers $[P(v,h)]$ If we take the derivative of the log-probability of the training data with respect to $W$ we get Equation (4).

$$\sum_{n=1}^{N} \frac{\partial \log(P(v^n))}{\partial w_{ij}} = \langle v_i | h_j \rangle_{data} - \langle v_i | h_j \rangle_{model} \tag{4}$$

here, $\langle v_i | h_j \rangle$ is the model distribution or the expected values of the data. Equation (5) is the learning rule for the weights of the network on log-probability-based training data.

$$\Delta w_{ij} = \varepsilon \left( \langle v_i | h_j \rangle_{data} - \langle v_i | h_j \rangle_{model} \right) \tag{5}$$

here, $\varepsilon$ is the learning rate.

In addition, activations of hidden or visible units can be conditionally dependent on visible or hidden units; The conditional probability of $h$ depending on $v$ is defined as in Equations (6)–(9).

$$P(h|v) = \prod_j (h_j|v) \quad h_j \in \{1,0\} \tag{6}$$

$$P(h_j = 1|v) = \sigma \left( b_j + \sum_i v_i W_{ij} \right) \tag{7}$$

$$P(v_i = 1|h) = \sigma \left( a_i + \sum_j h_j W_{ij} \right) \tag{8}$$

$$\sigma(x) = \left( 1 + e^{-x} \right)^{-1} \tag{9}$$

### 3.2. Recurrent Neural Networks

RNNs are a type of artificial neural network in which the connections between units form a directed loop. RNNs are feedforward neural networks with edges that expand along with adjacent time steps, introducing the idea of time to the traditional neural network paradigm [24]. The difference between RNN and the feedforward networks, NNs do not have traditional edge-to-edge loops. However, edges connecting adjacent time steps, called repeating edges, can form loops, including loops of length that are self-connecting from a node [25]. Nodes with recurring edges at time t are taken their input from the hidden node values $h^{t-1}$ in the network prior state and the present data point $x^t$ as shown in Equation (10). The output $y^t$ is calculated at time $t$ by giving the values $h^t$ of the hidden node at time t as shown in Equation (11). The input $x^{t-1}$ at time $t-1$ can affect the output at time $t$ through recursive connections to $y^t$ and beyond [26].

$$h^t = \sigma \left( W^{hx} x^t + W^{hh} h^t + b_h \right) \tag{10}$$

$$y^t = sortmax\left(W^{hx}h^t + b_y\right) \tag{11}$$

Here, $W^{hx}$ is input and the hidden layer conventional weights. $W^{hh}$ is the recurring weights matrix of the hidden layer and itself in adjacent time steps. The bias parameters vectors are $b_h$ and $b_y$. An illustration of a simple RNN structure is given in Figure 2.

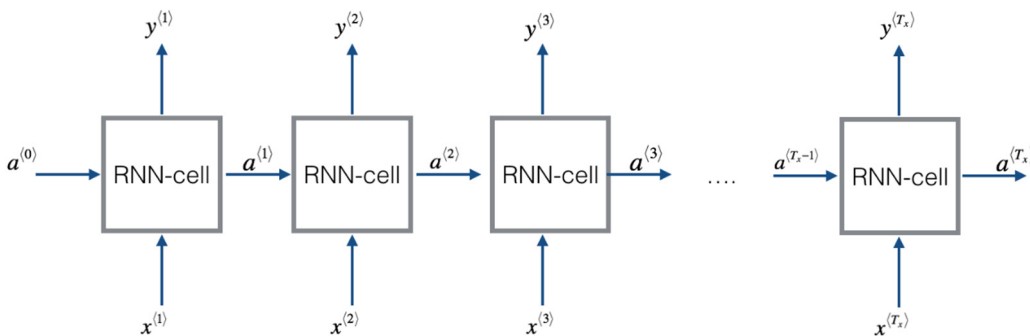

**Figure 2.** RNN Structure.

### 3.3. RNN-RBM Model

RNN and RBM DL methods can predict a temporal sequence. Combining each one of them into a deep architecture called RNN-RBM can take advantage of both models for capturing complex temporal dependencies in the data. This architecture is obtained by combining and including RBM at every time step with RNN, therefore the RBM model parameters are determined from an RNN as shown in Figure 3 [19].

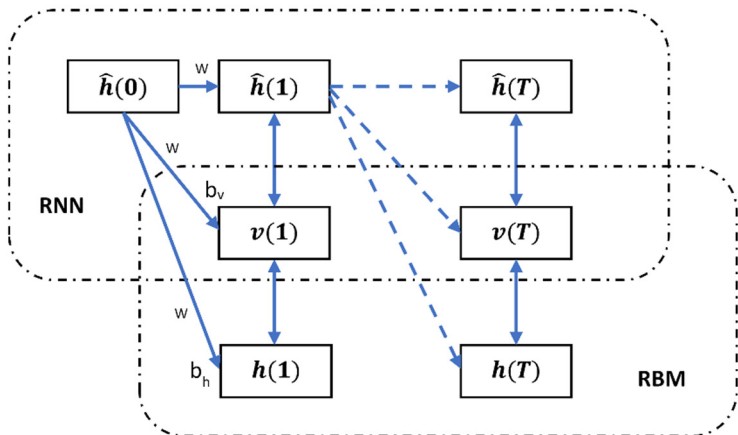

**Figure 3.** The structure of RNN-RBM.

The hidden units of RNN layers at time t are connected to each other in chain configuration and connected with visible layers of RBM as shown in Equation (12) [27].

$$\hat{h}^t = \sigma\left(W_2 v^t + W_3 \hat{h}^{t-1} + b_{\hat{h}}\right) \tag{12}$$

where $\hat{h}^t$, $W_2.W_3$ and $b_{\hat{h}}$ are the single-layer RNN-RBM parameters.

### 3.4. Message Queuing Telemetry Transport

Message queuing telemetry transport MQTT is an application layer protocol that runs on top of the TCP/IP transport layer protocol for data transfer. It is a protocol suitable for resource-constrained devices and was developed by Andy Stanford-Clark and Arlen Nipper in 1999. It is a lightweight and easy-to-use protocol, which provides a convenient transfer facility for communicating in resource-constrained systems such as the IoT. The design intent of MQTT is to provide reliable message transmission in environments such

as low bandwidth and unreliable networks for resource-constrained systems. The delivery of data is done using client-server and broadcast-subscriber mechanisms [28].

## 4. The Proposed Architecture

The proposed system architecture for cloud-based blood glucose level perdition IoT device is shown in Figure 4. The system consists of three main parts IoT device with a blood glucose sensor and display, an MQTT broker to provide the connection between the IoT device and the cloud environment, and cloud-based perdition RNN-RBM DL model. The IoT device includes a glucose level sensor to measure and monitor the patient's blood glucose level values and send it to the cloud environment via Wi-Fi using the MQTT protocol. The cloud environment contains a database to store the patient's glucose levels for the 20 data points in the last 100 min which then feed to the DL model used to predict the glucose level up to 30 min horizon into the future. The predicted value of glucose level is sent back to the IoT device to display to the patient.

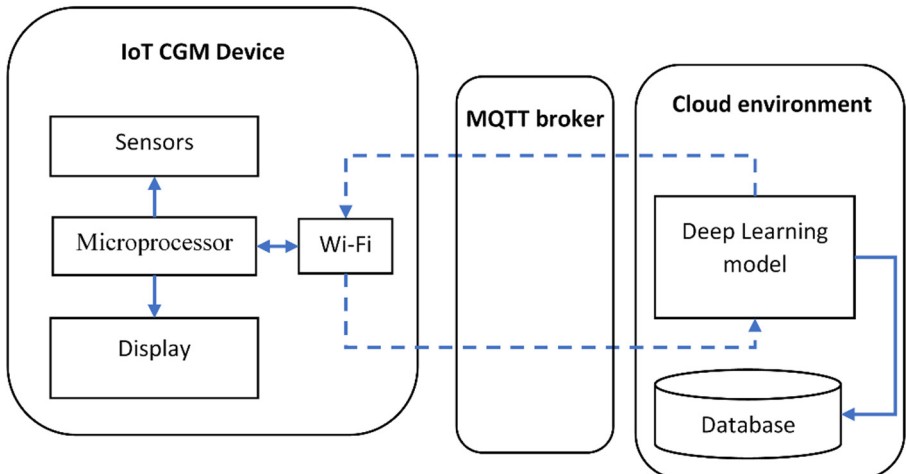

**Figure 4.** The proposed system architecture.

## 5. Experimental Evaluation

The dataset used to train the proposed RNN-RBM DL model for blood glucose level prediction is obtained from Diabetes Research in Children Network (DirecNet). The dataset contains historical data of blood glucose levels reordered in 5 min intervals for 110 T1D patients with ages between 7–17. The data were collected and obtained based on patients' and parents' approval in 3 months period and in separated sessions to ensure patient comfort and safety [29]. Table 1 summarizes information about the considered patients, where Record ID represents the sample number for taking glucose level measurements for each patient with a specific Patient ID.

**Table 1.** Dataset Information.

| Record ID | Patient ID | Reading Date | Reading Time | Sensor Glucose Level mg/dL |
|---|---|---|---|---|
| 35 | 2 | 1/1/2000 0:00 | 10:12 p.m. | 164 |
| 36 | 2 | 1/1/2000 0:00 | 10:17 p.m. | 160 |
| 37 | 2 | 1/1/2000 0:00 | 10:22 p.m. | 152 |
| 38 | 2 | 1/1/2000 0:00 | 10:27 p.m. | 145 |
| 39 | 2 | 1/1/2000 0:00 | 10:32 p.m. | 135 |
| 40 | 2 | 1/1/2000 0:00 | 10:37 p.m. | 128 |
| 41 | 2 | 1/1/2000 0:00 | 10:42 p.m. | 125 |
| 42 | 2 | 1/1/2000 0:00 | 10:47 p.m. | 118 |
| 43 | 2 | 1/1/2000 0:00 | 10:52 p.m. | 116 |

In this work, a subset of randomly selected 10 patients with glucose level data points more than 1000 is considered as shown in Table 2 below.

**Table 2.** The information of the 10 randomly selected patients.

| Patient ID | Glucose Level Measurement Data Points |
|---|---|
| 1 | 1053 |
| 5 | 1139 |
| 24 | 1589 |
| 45 | 1425 |
| 60 | 1165 |
| 71 | 1090 |
| 80 | 1055 |
| 85 | 1671 |
| 97 | 1448 |
| 104 | 1303 |

For each one of the selected patients, 80% of the glucose level values are considered for training the proposed RNN-RBM DL model and the remaining 20% is considered for testing and evaluating the model performance.

To avoid model overfitting issues a 10- fold Blocked cross-validation method [30] is considered for performance evaluation of the model.

The hyperparameters selected of the proposed RNN-RBM DL method for blood glucose level prediction are shown in Table 3.

**Table 3.** Model hyperparameter.

| Hyperparameter | Value | Search Space |
|---|---|---|
| Number of hidden units | 100 | (50–15) |
| Window size | 20 data points | (10–30) |
| optimizer | Adam | (Adam, SGD) |
| Batch size | 512 | (64–1024) |
| Learning rate | 0.001 | (0.001–0.1) |
| epochs | 150 | (50–250) |

The best values of the hyperparameters are selected through an experimental method based on the search space value range shown in Table 3.

In our evaluation experiments, the presented model is trained using batch learning data are sent to the network as batches determined by a sliding window.

The DL model is trained on the cloud environment using the training data and evaluated using the test data for each one of the selected 10 patients mentioned above. Root mean square error RMSE and mean absolute error MAE metrics are used to evaluate the performance of the model, RMSE and MAE can be calculated based on Equations (13) and (14), respectively [31].

$$RMSE = \sqrt{\frac{\sum_{i=1}^{n}(predicted_i\_actual_i)^2}{n}} \qquad (13)$$

$$MAE = \frac{1}{n}\sum_{i=1}^{n}|predicted_i\_actual_i| \qquad (14)$$

where *n* represents the number of samples.

Figure 5 shows the RMSE for the proposed DL model during training and validation in different epochs.

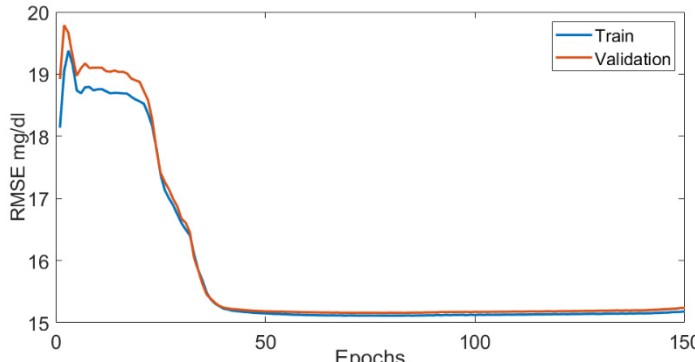

**Figure 5.** The RMSE results of the proposed model in different epochs.

Table 4 shows the evaluation results of the RNN-RBM DL model for glucose blood level perdition compared to the regular RNN method using the data of the selected 10 patients.

**Table 4.** The evaluation results.

| Patient Number | Patient ID | RNN-RBM Method | | RNN Method | |
|---|---|---|---|---|---|
| | | RMSE | MAE | RMSE | MAE |
| 1 | 1 | 15.589 | 13.185 | 18.617 | 13.319 |
| 2 | 5 | 17.962 | 15.114 | 19.165 | 15.238 |
| 3 | 24 | 15.747 | 14.352 | 21.429 | 13.318 |
| 4 | 45 | 15.278 | 11.990 | 18.485 | 13.317 |
| 5 | 60 | 15.245 | 13.920 | 18.559 | 13.319 |
| 6 | 71 | 14.539 | 10.857 | 16.594 | 11.646 |
| 7 | 80 | 15.589 | 12.684 | 18.720 | 12.454 |
| 8 | 85 | 14.356 | 10.703 | 17.543 | 13.868 |
| 9 | 97 | 15.139 | 12.226 | 16.545 | 13.626 |
| 10 | 104 | 16.446 | 14.229 | 20.523 | 13.085 |
| Average for all Patients | | 15.589 | 12.926 | 18.618 | 13.319 |

Based on the results shown in Table 4 the proposed RNN-RBM achieved an average RMSE value of 15.589 and 12.926 of MAE for all 10 patients' evaluation data, where the RNN method achieved an average of 18.618 of RMSE and 13.319 of MAE. Therefore RNN-RBM has outperformed the regular RNN model for glucose level prediction in terms of error since it has 16.27% less error.

Figure 6 shows the predictions from the proposed RNN-RBM model with respect to the actual value.

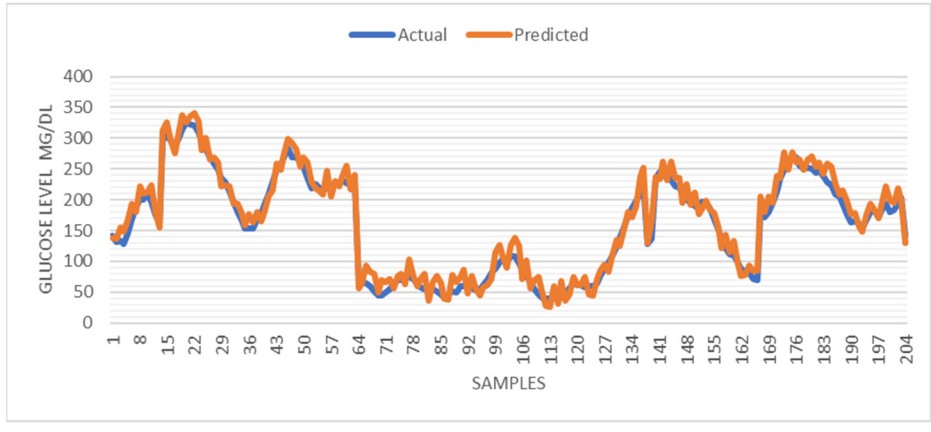

**Figure 6.** The predicted glucose level using the RNN-RBM model versus the actual level.

Table 5 shows a comparison between the presented RNN-RBM method for blood glucose level prediction and similar related SoTA works with regard to RMSE accuracy.

**Table 5.** Comparison between the proposed RNN-RBM method results and the other well-established methods in the literature.

| Method | RMSE | Dataset | Number of Patients | Model Parameters |
|---|---|---|---|---|
| RNN [7] | 18.867 | OhioT1DM | 6 | Layers 256 learning rate 0.001 batch size 1024 Adam optimizer 200 epochs Length of sequences 12 |
| DRNN [8] | 27.4 | OhioT1DM | 6 | Hidden nodes 32 Batch size 512 learning rate 0.001 |
| JNN [9] | 18.4 | Unit of Endocrinology and Diabetology of Campus Bio-Medico University Hospital | 17 | Hidden neuro 4 |
| LSTM [10] | 21.747 | GoCARB | 26 | Layers 4, 8, 64, 8 1300 epochs |
| CNN [12] | 19.2 | OhioT1DM | 6 | Layers 5, 64, 32 window of size 16 |
| LSTM [13] | 37.8 | OhioT1DM | 5 | Layers 3 Learning rate 0.0001 5000 epoches batch size of 500 |
| ARX [15] | 19.53 | OhioT1DM | 6 | Alpha = 0.0001 window size 576 |
| VanillaLSTM [15] | 19.58 | OhioT1DM | 6 | Layers 3, 10 10 epochs Adam optimization learning rate 0.001 window size to 576 |
| The proposed RNN-RBM | 15.589 | DirecNet | 10 | Table 3 |

The comparison results in Table 5 show that the proposed RNN-RBM method achieves the lowest value of RMSE and is considered the best method in the SoTA according to accuracy.

## 6. Conclusions

In this paper, a cloud-based DL model based on cascaded RNN and RBM methods is proposed for wearable IoT continuous glucose monitoring CGM devices. The proposed Cloud&DL-based model is used to predict the glucose levels in the blood for 30 min horizon in 5 min intervals ahead based on the prior 20 samples of blood glucose levels of the patient. Therefore, the model is able to predict the fluctuations in glucose level that occurred in 5 min intervals which are sufficient for the patient to take any action in case of perdition a hyperglycemia condition.

The DL model is implemented and trained using batch learning in the considered Cloud computing environment which provides many features such as virtualization and resource sharing. Cloud computing also provides machine learning specific services such as Machine Learning as a Service (MLaaS) which is specifically designed for machine learning perdition applications to provide services to multiple users simultaneously via using multiple virtual machines.

For the actual application, the model can be initially pre-trained using batch learning and since the model is implemented in a cloud environment online learning can be used for continuously updating the model to learn from patient data.

Lightweight MQTT communication protocol is considered for data exchanging between the low-power CGM IoT device and the cloud-based prediction model. Real data of 10 T1D patients were used to train and evaluate the proposed DL RNN-RBM method. From the experimental results, it was demonstrated that the RNN-RBM DL model acquired the highest performance regarding accuracy prediction which outperformed the regular RNN methods and similar ones in the state-of-the-art. Since the model is implemented in a Cloud computing environment it required a high QoS service provider with a high level of security for patients' information security and privacy.

For future research, the proposed architecture can extend to be used with insulin delivery devices to provide the patients with the required amount of insulin based on the predicted future blood glucose levels. In this case, a second deep learning model can be deployed to estimate the required amount of insulin that should provide to the patient based on the future values of blood glucose levels obtained from the presented model in this work. The estimated amount of insulin then will be supplied to the patient via the insulin delivery device. Furthermore, optimization methods can be used for finding the optimal models hyperparameters instead of the traditional experimental method.

**Author Contributions:** Conceptualization, A.R.N., L.A. and A.J.H.; methodology, A.R.N. and A.J.H.; software, A.R.N., M.A.F.; validation, A.R.N., A.M.H., L.A., J.S., Y.D. and A.J.H.; formal analysis, A.R.N., L.A.; investigation, A.R.N.; resources, A.R.N.; data curation, A.R.N.; writing—original draft preparation, A.R.N., A.J.H. and L.A.; writing—review and editing, A.R.N., A.M.H., A.J.H., A.A., L.A., M.A.F., J.S. and Y.D.; visualization, A.R.N. and M.A.F.; supervision, A.J.H.; project administration, A.J.H., J.S. and Y.D.; funding acquisition, A.J.H., A.A., L.A. All authors have read and agreed to the published version of the manuscript.

**Funding:** This research received no external funding.

**Conflicts of Interest:** The authors declare no conflict of interest.

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
