# Peer review of "IoT and Cloud Computing in Health-Care: A New Wearable Device and Cloud-Based Deep Learning Algorithm for Monitoring of Diabetes"

_electronics, doi:10.3390/electronics10212719_

Round 1

Reviewer 1 Report

In this paper, the authors have proposed a cloud and deep learning based wearable system for prediction of blood glucose levels. The paper is well-written, but I have the following comments:

  1. Specific RNN models like LSTM and also HTM have shown better promise in prediction of time-series data. Why haven’t the authors use those? The authors need to provide sufficient motivation behind the methods used by them in this paper.
  2. The description of the methods like RBM and RNN is well-known in the literature. The authors should cite relevant papers rather than describing them in detail as they don’t provide any scientific contributions by the authors.
  3. The motivation behind combining the RNN and RBM model is not convincing. Combining two models is not always straightforward. The authors mention that by this they can take advantages of both models which is not always true as the drawbacks of each model can also combine.
  4. Table 1 dataset information is tough to understand. What is record ID?
  5. Does the training and the testing sample include the data from all the 110 patients? The experimental evaluation section is incomplete. Please provide all the relevant information.
  6. Can the model predict sudden events or sudden fluctuations in blood glucose level? Most models can perform good prediction if the time-series data is normal.
  7. How can the authors determine that the accuracy of the proposed model is not due to the overfitting of data as 80% of the data is used for training and 20% for testing? Overall, the scientific contribution of this paper is not significant.

Author Response

Dear Reviewer 

Reviewer 2 Report

The paper and the topic are interesting and relevant. I find the Introduction, Related work and Proposed approach good, but the remainder of the paper could be improved.

In particular, there is more information about the data missing. How long does it span? Please give a summary of the study participants.

How were the hyper parameters tuned?

As far as I understand, you are using 4 datapoint (20 minutes of data collected every 5 minutes) to predict the next 6 datapoint (30 minutes into the future). Is this correct? If yes, is 4 datapoint sufficient? Why not use more? You talk about long effect on glucose level, and impact of poor sleep etc. Using 20 minutes of data does hardly capture that. Please elaborate.

How were the methods in Table 4 trained and tuned? Please explain. The conclusion needs to be more elaborate, with limitations and plan for future work.

Is the methodology setup for online or batch learning? Since data is being streamed and the model trained on the cloud, I suspect it could be online, which would also be ideal. But this is not clear to me.

The authors should motivate their choice of the choice of RNN-RBM model better. 

Author Response

Dear Reviewer 

Reviewer 3 Report

The article is interesting. It deals with important aspects. Any scientific/technical proposal that can help protect health or save lives deserves attention. Nowadays, a wide spectrum of AI is used for prediction purposes in health care. Therefore, at the end of section 2, the authors should clearly state why their solution is better than existing solutions and what problem, drawback, shortcoming in other solutions it solves.
It should be stated whether the learning process was done once, does the system take into account re-learning from new data? If so, is this likely to affect ongoing patient monitoring?
Please provide information on how the issue of parallelization of data processing for multiple patients has been resolved? Are there any limitations in this matter?
It is required to present the actual implementation of the system used by the authors, i.e. what measurement devices were used, how the data from patients were transmitted, were there any problems with losses of transmitted data?

Author Response

Dear Reviewer 

Round 2

Reviewer 1 Report

The authors have successfully addressed the comments of the reviewers. The quality of the paper has improved after the revision and I recommend it for publication.

Author Response

We would like to thank the reviewer for his/her time and valuable comments. 

Reviewer 3 Report

After reviewing the authors' responses and the additions made, I still suggest expanding the information on further research in conclusion. Please also indicate clearly whether the medical data used required the consent of the patients and whether they were obtained in an ethically unquestionable manner. 

Author Response

Dear Reviewer 
